# Relationship between health literacy and attitudes toward acupuncture: A web-based cross-sectional survey with a panel of Japanese residents

**Yuse Okawa**[1,2]*, **Norio Ideguchi**[1], **Hitoshi Yamashita**[1,2]*

**1** Graduate School of Health Sciences, Morinomiya University of Medical Sciences, Osaka, Japan,
**2** Morinomiya University of Medical Sciences Acupuncture Information Center, Osaka, Japan

* yuse_okawa@morinomiya-u.ac.jp (YO); yamashita@morinomiya-u.ac.jp (HY)

**Data Availability Statement:** Raw data files used in this study are available from the figshare.(https://doi.org/10.6084/m9.figshare.21368898.v4).

## Abstract

The relationship between health literacy of Japanese people, their attitudes toward acupuncture, and their behavior in choosing this therapy is currently unclear. Therefore, for this study, we conducted a web-based survey to address this unknown relationship. A questionnaire comprising four categories (health status, health literacy, previous acupuncture experience, and attitudes toward acupuncture) was administered to 1,600 Japanese participants. For this study, we performed cross-tabulation and path analysis to examine the relationship between each questionnaire item. The mean score of participants' health literacy was 3.41 (SD = 0.74), and older, educated, female participants tended to have higher health literacy. The respondents perceived acupuncture to be effective for chronic low back pain, tension-type headache, and knee pain due to osteoarthritis (40.0%, 38.7%, and 21.8%, respectively). Contrastingly, acupuncture was perceived as far less effective for postoperative nausea/vomiting and prostatitis symptoms (8.3% and 8.7%, respectively). Of the total study respondents, 34.4% reported that they would try acupuncture only if recommended by clinical practice guidelines, and 35.6% agreed that acupuncture is safe. The path analysis showed that attitudes toward acupuncture were significantly influenced by the participants' health literacy, number of information sources, and previous acupuncture experience. However, it was also found that experience with acupuncture was not directly associated with health literacy. Although the Japanese population with higher health literacy is more likely to perceive acupuncture positively, they do not necessarily have sufficient relevant knowledge of the clinical evidence. Therefore, their decision to receive acupuncture may be more dependent on personal narratives rather than clinical evidence. Thus, future challenges lie in individual education of the population on how to choose a reliable health information source, and organizational efforts to provide more reliable health information.

**Funding:** This research was funded by the 2018 President's Incentive Research Grant from Morinomiya University of Medical Sciences (YO). The funder had no role in the study design, data collection, analysis, decision to publish, or manuscript preparation.

**Competing interests:** The authors have declared that no competing interests exist.

## Introduction

According to The World Health Organization (WHO)'s definition, health literacy refers to cognitive and social skills which determine the motivation and ability of individuals to gain access to, understand and use information in ways which promote and maintain good health [1]. The WHO identifies health literacy as one of the three pillars of health promotion (good governance, healthy cities and health literacy) and recognizes it as a critical determinant of health [2]. Previous studies suggest that low health literacy is associated with poorer health outcomes and higher costs, such as more hospitalizations, greater use of emergency services and higher mortality rates in older people [3–5]. Studies including measurement of health literacy have been conducted, particularly in the United States, although the evidence on the relationship between health literacy and health disparities is still scarce and limited [6].

In Japan, health literacy scales with good reliability and validity have already been developed or translated by previous studies, and Japanese people's health literacy has been assessed using those scales [7–9]. It has been reported that Japanese people with higher health literacy are more likely to obtain sufficient health information from multiple sources and, thus, are less likely to engage in risky habits like smoking and regular drinking [10]. However, a survey has found that Japanese health literacy is relatively lower than that of Europeans [9]. The authors who conducted that survey speculate that this may be due in part to an insufficient primary health care system and difficulty accessing reliable health information, but no solid factors have been identified. Moreover, it is unknown whether the health literacy of the Japanese people influences the use of healthcare measures for which the evidence is controversial and not well established. Representative of such healthcare measures would be the area of healthcare grouped as complementary therapies [11, 12]. Studies in the United States suggest that people with high health literacy tend to use complementary therapies at a higher rate [13, 14]. In contrast, to our knowledge, no such studies have been reported from East Asian countries including Japan, China and South Korea.

One typical example of complementary therapies with insufficiently established evidence that has been used by some percentage of the Japanese population is acupuncture therapy. It has been reported that in Japan, approximately 5–7% of people receive acupuncture at least once a year [15, 16]. Musculoskeletal problems were the most common condition for which acupuncture was used (approximately 80%), and referral by family or friends was the most common reasons for people receiving acupuncture (approximately 60%) [16]. Around the 2000s, it was unknown to what extent information on the evidence for acupuncture was widespread within the Japanese population; additionally, it was unclear how much their health literacy played a role in their decision to use acupuncture for their intended condition. Incidentally, existing survey reports suggest that Japanese users of acupuncture tend to have lower education levels [15, 16], but these results were probably confounded by age factor because the older population, who were relatively less educated, were more likely to use acupuncture than the younger [16]. Moreover, the relationship between health literacy and education levels has been controversial among scholars in Japan [9, 10, 17].

Recently, studies have found some evidence of the clinical effectiveness and safety of acupuncture. Albeit insufficient, there are some positive conclusions regarding acupuncture for several health conditions like chronic low back pain and headaches in the Cochrane Database of Systematic Reviews [18–23]. Furthermore, several evidence-based clinical practice guidelines have been developed in Japan that include recommendations for acupuncture [24]. Under these circumstances, it remains unknown whether people with high health literacy adequately scrutinize the evidence, affirm acupuncture for the conditions for which evidence is shown, and consequently receive this treatment.

In this study, we conducted a web-based questionnaire survey in Japan to address the gap in this body of research. The aim of the study was to investigate the relationship between people's health literacy and their health information sources and their attitudes toward acupuncture. In addition, this study aimed to understand the interrelationships among multiple factors such as sociodemographic characteristics, health literacy, health status, information sources, attitudes toward acupuncture and experience of acupuncture. Finally, to the WHO policy which encourages member states to provide individuals with reliable information on the benefits and risks of integrating traditional and complementary medicine into their healthcare system [25], our study can contribute insights for further understanding of and research on relevant education measures for the public in terms of health literacy.

## Materials and methods

We referred to the STROBE checklist [26], CHERRIES checklist [27], and reporting guidelines for SEM (structural equation modeling) suggested by Morrison et al. [28] when reporting the current study (S1–S3 Tables).

### Study populations and procedures

For this study, we conducted a web-based cross-sectional questionnaire survey between January 27 and February 3, 2020. The survey was commissioned to Mellinks Co., Ltd., Tokyo, Japan (https://www.mellinks.co.jp/), an internet research company that has nationwide panels by age group.

This was a closed, panel-based survey that required registration for eligible respondents to be able to complete it. It was conducted on the Mellinks website for monitors registered with Mellinks' affiliated research firm. Prior to the questionnaire survey, approximately 40,000 potential survey monitors were randomly selected and contacted via e-mail or web notification with the survey protocol. The protocol notice provided an overview of the study design, including the name of the study investigator, on a webpage, and confirmed the participants' willingness to participate in the study. Those who completed this survey were compensated; however, the exact amount of compensation was not disclosed by the research firm that was affiliated with Mellinks. Incidentally, the typical compensation for such surveys ranges from a few yen to 10 yen for the preliminary survey and from 50 yen to 100 yen for the actual survey.

Those who agreed to participate in this study were asked to provide their educational attainment and occupation, which indicated the completion of the preliminary survey. Thereafter, Mellinks notified the consenting participants via e-mail or web-based invitation, before commencing the survey. All 11 questions were displayed on a single web page, and the participants were asked to answer all. To prevent missing data, this question session was designed in a manner that it will be considered as incomplete if there were unanswered questions. Once the participants had answered all the questions, they were asked to review their answers again and were allowed to revise them, if required. The survey targeted 160 men and 160 women in each age group (20s (20–29 years), 30s (30–39 years), 40s (40–49 years), 50s (50–59 years) and 60s (60–69 years)).

Thus, a total of 1,600 participants living in Japan were included in this study. When 160 respondents in each age and sex group responded, the respective quota was closed. Responses were tabulated automatically.

### Development of questionnaire

We developed a questionnaire comprising four categories (S4 Table): health status (Q1, 2), health literacy (Q3, 4), experience of receiving acupuncture (Q5, 6), and recognition and

choice behavior regarding acupuncture (Q7–11). Regarding the health literacy measurement of Q3, we used a 5-item questionnaire developed by Ishikawa et al., who are considered to be specialists in public health who conducted a pilot study questioning 190 male office workers during an annual health check-up. The scale items of the questionnaire draft were constructed to directly reflect the WHO definitions [7]. Each item was rated on a 5-point scale, ranging from 1 (strongly disagree) to 5 (strongly agree). We chose this scale [7] for this study because of the following reasons: the internal reliability of the scale has been evaluated and warranted (Cronbach's α = 0.86); it comprises only five questions, making it compact and user-friendly for respondents; it was subsequently used in some surveys targeting a bigger sample of the Japanese population [29, 30].

As for the recognition and choice behavior of acupuncture, the questionnaire asked for a reliable information source (Q7), expected health conditions that may benefit from acupuncture (Q8), possible influence of clinical practice guidelines on the decision to receive acupuncture (Q9), and the safety of acupuncture (Q10,11). For Q8–10, we used a 5-point rating scale similar to Q3. For Q8 ("Do you think that acupuncture is effective for the following symptoms or diseases?"), we selected six conditions for which the Cochrane Database of Systematic Reviews concluded positively on the clinical benefit of acupuncture as of January 2020 [19–23, 31].

Along with the abovementioned 11 questions, we created questions for basic information on sociodemographic attributes like sex, age, educational attainment, occupation, and residential area. The developed questionnaire draft was pre-tested using our university staff members who were not included in the study panel to make it easier to answer. Based on the inconvenience pointed out by several of the staff members, we improved the questionnaire to arrive at the final version. Thereafter, the completed questionnaire was arranged by Mellinks for the web survey. We checked the usability of the survey screen before releasing it to the participants.

## Sample size calculation

The lifetime use of acupuncture in Japan has been reported to be approximately 25% [16]. Assuming a confidence level of 95%, an acceptable sampling error of 5%, and a response rate of 50%, 384 samples were required for that 25% in this study (n = $1.96^2$ x 0.5(1–0.5)/$0.05^2$). Therefore, we decided to collect 1,600 samples to ensure reliability.

## Data analysis

For data analysis, descriptive statistics, like frequencies, means, and cross-tabulations, were calculated. Based on responses to Q3 (health literacy measures), respondents were divided into two groups of health literacy according to the median score of five items (5-point rating each): a median score of four or more was regarded as the higher health literacy (HHL) group, and that of less than four, as the lower health literacy (LHL) group [7]. For Q8–10 which assessed the participants' attitude toward acupuncture, respondents were divided into two groups of affirmation on acupuncture based on each respondent's 5-point ratings; a score of four or more was regarded as the acupuncture approval (AA) group and that of less than four as the acupuncture disapproval/Neutral (AD/N) group. In Q8, the division of the respondents into two groups (AA and AD/N group) depended on whether their median score for six symptoms was four or higher.

Using these classifications, we assessed the interrelationships among sex, health literacy, experience of acupuncture, and attitudes toward acupuncture through cross-tabulation. Pearson's chi-squared test was used for these analyses. Odds ratios and 95% confidence intervals were shown as effect sizes.

Additionally, we performed a path analysis to examine direct and indirect interrelationships among sociodemographic factors (sex, age, and educational attainment), health literacy, the number of information sources, health status, and experience and attitudes toward acupuncture among the participants. For health literacy, we used continuous variables, while dichotomized data were used for cross-tabulation. Model parameters were estimated using the maximum-likelihood estimation. The estimation was performed by assuming endogenous correlations. For this, first we performed a path analysis to test the hypothesis model established based on the results of above cross-tabulation and previous studies suggesting a relationship between health literacy and several factors, like sociodemographic characteristics or the number of information sources [10], then trimmed non-significant paths to reach a final model. The fitness of the model was evaluated using comparative fit index (CFI) and root mean square error of approximation (RMSEA). If the CFI was larger than 0.95 and the RMSEA was < 0.05, the model was considered to be acceptable [32].

All statistical analyses were performed using the software, Jamovi Version 2.3.0 [33]. The analysis was conducted without weighting the sample. For path analysis, we used Jamovi's modules of PATH ANALYSIS 0.8.0. The significance level was set at $p < 0.05$.

### Ethical statement

The study protocol was approved by the Ethics Committee of Morinomiya University of Medical Sciences (No. 2019–100). Regarding the protection of personal information, we confirmed that the internet research company, Mellinks, obtained PrivacyMark [34]. The research panel members were informed of the objective, content, approximate time required for response, and protection policy of personal information, and the fact that participation was not forced prior to starting the questionnaire. When the panel members agreed to become respondents with a written document on the web page, they clicked on an agreement button, which started the questionnaire survey. The authors had no special access privileges to the data that could identify individual respondents.

## Results

The total number of those who agreed to participate at the pre-survey stage was 3,292. Among them, the collection of responses continued until the number of respondents reached 160 in each of the age and sex quotas, and a total of 1,600 responses were finally collected. Therefore, the completion rate was 48.6%.

### Sociodemographic characteristics of respondents

The characteristics of the respondents in terms of sex, age group, educational attainment, occupation, and residential area are shown in Table 1. Among the male respondents, university graduates and office workers were the majority in the sample, while high school graduates and houseworkers were the majority among females. It is noteworthy that the number of respondents in each age group was 160 for both males and females and was not proportional to the population structure of Japan.

### Health status (Q1, 2)

In total, the proportion of those who thought they were healthy ("agree" + "strongly agree"), unhealthy ("disagree" + "strongly disagree") and "neither" was 45.3%, 28.9% and 25.9%, respectively. This trend could also be seen in the 60s age group (46.6%, 30.0%, and 23.4%, respectively). Regarding the conditions or diseases being treated currently, 485 respondents

**Table 1. Sociodemographic characteristics of the respondents.**

| | Total (n = 1,600) | Male | Female |
|---|---|---|---|
| | n (%) | 800 (50.0) | 800 (50.0) |
| Age group (in years) | | | |
| 20s (20–29) | 320 (20.0) | 160 (10.0) | 160 (10.0) |
| 30s (30–39) | 320 (20.0) | 160 (10.0) | 160 (10.0) |
| 40s (40–49) | 320 (20.0) | 160 (10.0) | 160 (10.0) |
| 50s (50–59) | 320 (20.0) | 160 (10.0) | 160 (10.0) |
| 60s (60–69) | 320 (20.0) | 160 (10.0) | 160 (10.0) |
| Educational attainment | | | |
| Junior high school | 39 (2.4) | 22 (1.4) | 17 (1.1) |
| High school | 470 (29.4) | 214 (13.4) | 256 (16.0) |
| Vocational school | 204 (12.8) | 81 (5.1) | 123 (7.7) |
| Junior college/ Higher professional school | 194 (12.1) | 42 (2.6) | 152 (9.5) |
| University | 624 (39.0) | 392 (24.5) | 232 (14.5) |
| Graduate school | 69 (4.3) | 49 (3.1) | 20 (1.3) |
| Occupation | | | |
| Self-employed | 98 (6.1) | 80 (5.0) | 18 (1.1) |
| Healthcare worker | 62 (3.9) | 20 (1.3) | 42 (2.6) |
| Office worker | 620 (38.8) | 440 (27.5) | 180 (11.3) |
| Public officer | 62 (3.9) | 48 (3.0) | 14 (0.9) |
| House worker | 253 (15.8) | 5 (0.3) | 248 (15.5) |
| Student | 48 (3.0) | 25 (1.6) | 23 (1.4) |
| Part-time worker | 238 (14.9) | 55 (3.4) | 183 (11.4) |
| Unemployed | 207 (12.9) | 120 (7.5) | 87 (5.4) |
| Other | 20 (1.3) | 9 (0.6) | 11 (0.7) |
| Residential area | | | |
| Hokkaido region | 99 (6.2) | 52 (3.3) | 47 (2.9) |
| Tohoku region | 93 (5.8) | 45 (2.8) | 48 (3.0) |
| Kanto region | 614 (38.4) | 329 (20.6) | 285 (17.8) |
| Chubu region | 238 (14.9) | 104 (6.5) | 134 (8.4) |
| Kinki region | 312 (19.5) | 161 (10.1) | 151 (9.4) |
| Chugoku region | 87 (5.4) | 48 (3.0) | 39 (2.4) |
| Shikoku region | 39 (2.4) | 15 (0.9) | 24 (1.5) |
| Kyushu region | 118 (7.4) | 46 (2.9) | 72 (4.5) |

(30.3% of the total 1,600 respondents) described at least one specific condition or disease. As per the classification by ICD-10, the most common diseases were endocrine, nutritional, and metabolic diseases (6.3%), followed by the circulatory system (6.1%), musculoskeletal system and connective tissue (4.8%), mental and behavioral disorders (4.5%), digestive system (4.2%), respiratory system (2.9%), and others in decreasing order.

## Health literacy (Q3, 4)

The mean score for the five items in Q3 (each item scored on a 5-point scale) was 3.41 (SD = 0.74), which tended to be higher with increasing age group. Divided into two groups (HHL ≥ 4, LHL < 4) according to the median score of five items, the number of respondents with HHL was 797 (49.8%) and those with LHL was 803 (50.2%). In the 20s age group, the proportion of male respondents with HHL tended to be small (Table 2).

**Table 2. Health literacy score and proportion of higher/lower health literacy by age and sex.**

| | Total (n = 1,600) | | | Male (n = 800) | | | Female (n = 800) | | | Chi-squared test |
|---|---|---|---|---|---|---|---|---|---|---|
| | HL score | HHL (≧4) | LHL (<4) | HL score | HHL (≧4) | LHL (<4) | HL score | HHL (≧4) | LHL (<4) | HHL LHL |
| | Mean (SD) | n (%) | n (%) | Mean (SD) | n (%) | n (%) | Mean (SD) | n (%) | n (%) | Male |
| | | | | | | | | | | Female |
| Age group (in years) | | | | | | | | | | |
| 20s (20–29) | 3.32 (0.77) | 141 (8.8) | 179 (11.2) | 3.22 (0.77) | 58 (7.3) | 102 (12.8) | 3.41 (0.75) | 83 (10.4) | 77 (9.6) | p = 0.005 |
| 30s (30–39) | 3.31 (0.77) | 139 (8.7) | 181 (11.3) | 3.31 (0.85) | 74 (9.3) | 86 (10.8) | 3.31 (0.67) | 65 (8.1) | 95 (11.9) | p = 0.310 |
| 40s (40–49) | 3.41 (0.78) | 154 (9.6) | 166 (10.4) | 3.42 (0.84) | 75 (9.4) | 85 (10.6) | 3.40 (0.71) | 79 (9.9) | 81 (10.1) | p = 0.654 |
| 50s (50–59) | 3.48 (0.66) | 172 (10.8) | 148 (9.3) | 3.46 (0.63) | 80 (10.0) | 80 (10.0) | 3.49 (0.69) | 92 (11.5) | 68 (8.5) | p = 0.178 |
| 60s (60–69) | 3.55 (0.68) | 191 (11.9) | 129 (8.1) | 3.52 (0.71) | 90 (11.3) | 70 (8.8) | 3.59 (0.64) | 101 (12.6) | 59 (7.4) | p = 0.210 |

HL: health literacy; HHL: higher health literacy; LHL: lower health literacy; SD: standard deviation

As responses to Q4, television was found to be the most common source of information followed by internet and blogs in all age groups, except for those in their 40s. Another feature was the use of social networking sites (SNS; Twitter, Facebook, Instagram) for obtaining relevant information, which decreased as the age group increased while medical doctors and newspapers showed the opposite trend (Fig 1).

Table 3 shows the relationship between the level of health literacy (HHL vs. LHL) and the use of each health information source. Significant differences in the use of health information

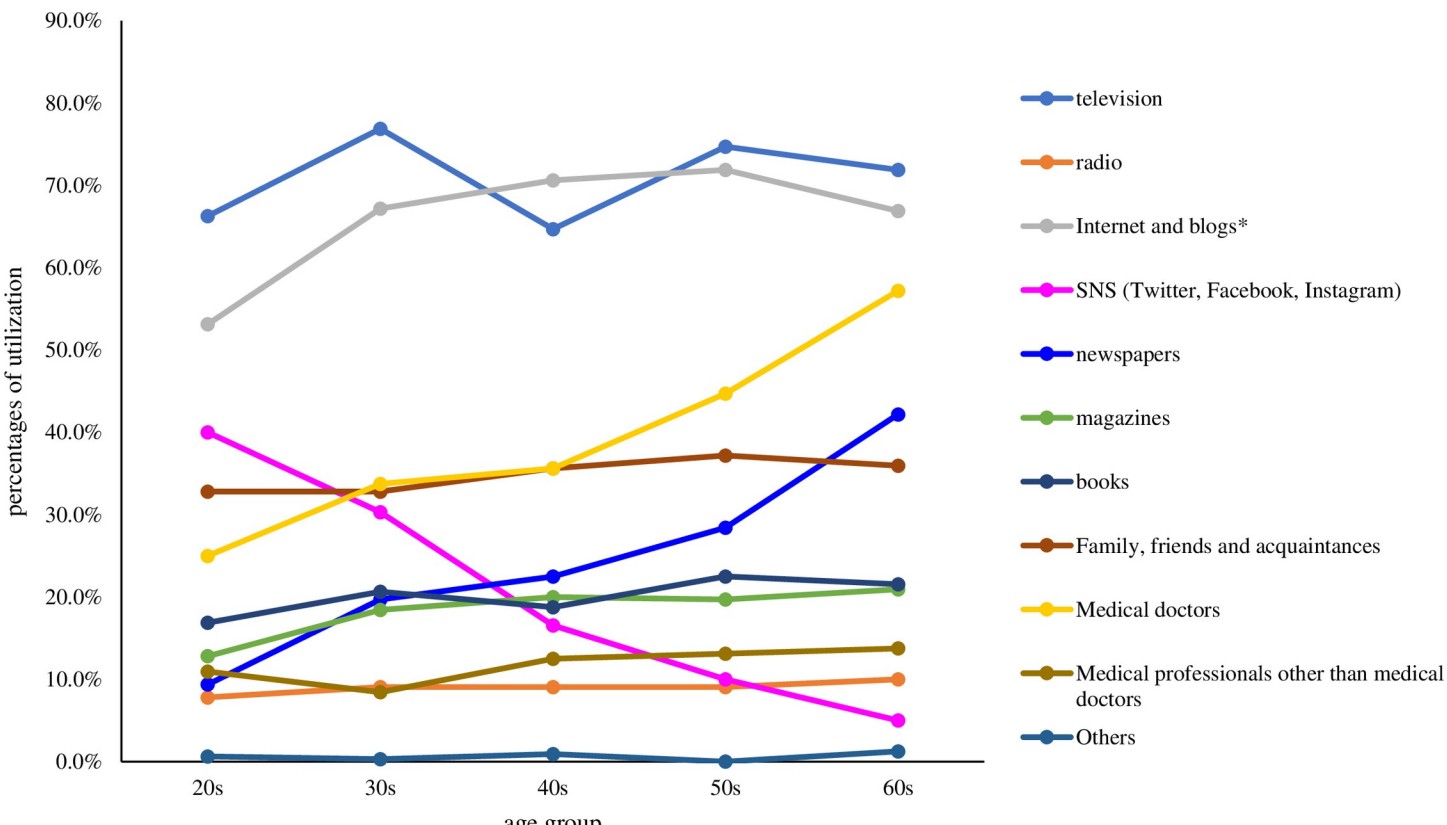

**Fig 1. Sources from which respondents in each age group get information about illness and health.** The respondents could answer with more than one response option; hence, this figure includes multiple answers. * Social networking sites (SNS) were excluded.

**Table 3. Relationship between level of health literacy and use of health information sources.**

| Information source | | Health literacy | | $\chi^2$ | df | p value* | OR | 95% CI |
|---|---|---|---|---|---|---|---|---|
| | | HHL | LHL | | | | | |
| Television | Yes | 592 | 542 | 8.91 | 1 | 0.003 | 1.39 | 1.12–1.73 |
| | No | 205 | 261 | | | | | |
| Radio | Yes | 75 | 69 | 0.33 | 1 | 0.568 | 1.11 | 0.78–1.56 |
| | No | 722 | 734 | | | | | |
| Internets and Blogs (except for SNS) | Yes | 622 | 433 | 103.61 | 1 | < .001 | 3.04 | 2.44–3.78 |
| | No | 175 | 370 | | | | | |
| SNS (Twitter, Facebook, Instagram) | Yes | 176 | 150 | 2.86 | 1 | 0.091 | 1.23 | 0.97–1.58 |
| | No | 621 | 653 | | | | | |
| Newspapers | Yes | 266 | 125 | 68.70 | 1 | < .001 | 2.72 | 2.14–3.46 |
| | No | 531 | 678 | | | | | |
| Magazines | Yes | 198 | 96 | 44.30 | 1 | < .001 | 2.43 | 1.86–3.18 |
| | No | 599 | 707 | | | | | |
| Books | Yes | 235 | 86 | 87.93 | 1 | < .001 | 3.49 | 2.66–4.57 |
| | No | 562 | 717 | | | | | |
| Family, Friends, and Acquaintances | Yes | 336 | 222 | 37.09 | 1 | < .001 | 1.91 | 1.55–2.35 |
| | No | 461 | 581 | | | | | |
| Medical doctors | Yes | 382 | 246 | 50.18 | 1 | < .001 | 2.08 | 1.70–2.56 |
| | No | 415 | 557 | | | | | |
| Medical professionals other than medical doctors | Yes | 130 | 58 | 31.86 | 1 | < .001 | 2.5 | 1.81–3.47 |
| | No | 667 | 745 | | | | | |

*Pearson's chi-squared test.

SNS: Social networking sites; HHL: higher health literacy; LHL: lower health literacy; df: degrees of freedom; OR: odds ratio; CI: confidence interval

sources were found according to the level of health literacy, except for radio and SNS. There was a particularly large odds ratio between HHL and LHL regarding the use of books, internet, and newspapers (3.49, 3.04, and 2.72, respectively).

## Experience of acupuncture (Q5, 6)

For the questions regarding experience of acupuncture, 61 respondents (3.8%) answered that they were currently receiving acupuncture treatment, and 70 (4.4%) answered that they were "not currently receiving but have received within the past year". Therefore, the percentage of annual use of acupuncture by the Japanese people was 8.2% (95% CI: 6.9–9.6). The percentage of those who had received acupuncture at least once in the past, that is, lifetime use, was 25.4% (95% CI: 23.3–27.6).

Among those who responded to "have never received" (1,194 respondents), the most common reason was because they "did not feel the necessity" (52.3%), followed by "because I was healthy" (25.0%), "because I thought it would be expensive" (15.6%), and "because acupuncture seemed painful" (14.2%). In the female respondents, the third most common reason was "because acupuncture seemed painful" while this answer was the sixth most common in the male respondents (Table 4).

## Information source, knowledge, and attitudes on acupuncture (Q7–11)

Regarding the source of information for deciding whether to receive acupuncture (Q7), the most common answer among all respondents was family, friends, and acquaintances (30.3%),

**Table 4. Experience of acupuncture.**

| Question number | Question | Total (n = 1,600) | Male (n = 800) | Female (n = 800) |
|---|---|---|---|---|
| | | n (%) | n (%) | n (%) |
| 5 | Have you ever received acupuncture and moxibustion treatment? | | | |
| | Yes | | | |
| | Currently receiving | 61 (3.8) | 37 (4.6) | 24 (3.0) |
| | Not currently receiving, but have received within the past year | 70 (4.4) | 35 (4.4) | 35 (4.4) |
| | Have not received within the past year, but have received in the past | 275 (17.2) | 147 (18.4) | 128 (16.0) |
| | No | | | |
| | Have never received | 1,194 (74.6) | 581 (72.6) | 613 (76.6) |
| | | Total (n = 1,194) | Male (n = 581) | Female (n = 613) |
| | | n (%) | n (%) | n (%) |
| 6 | If you answered "have never received" to Q5, please choose the reason from the following (multiple answers are possible) | | | |
| | Because I was healthy | 298 (25.0) | 162 (27.9) | 136 (22.2) |
| | Because I did not feel the necessity | 625 (52.3) | 309 (53.2) | 316 (51.5) |
| | Because I did not expect to feel the efficacy of acupuncture | 106 (8.9) | 64 (11.0) | 42 (6.9) |
| | Because acupuncture seemed painful | 170 (14.2) | 56 (9.6) | 114 (18.6) |
| | Because I thought acupuncture is a dubious treatment | 78 (6.5) | 41 (7.1) | 37 (6.0) |
| | Because I was afraid of the side effects | 41 (3.4) | 13 (2.2) | 28 (4.6) |
| | Because I thought it would be expensive | 186 (15.6) | 74 (12.7) | 112 (18.3) |
| | Because I did not know about acupuncture | 107 (9.0) | 60 (10.3) | 47 (7.7) |
| | Other | 16 (1.3) | 5 (0.9) | 11 (1.8) |

followed by internet and blogs (except for SNS) (29.7%), and medical doctors (27.0%). The respondents were most likely to use the internet and blogs in their 20s to 40s. In their 50s, they would essentially rely on family, friends, and acquaintances, and on medical doctors in their 60s (Fig 2).

The most common condition for which acupuncture was recognized as effective among the six conditions listed in Q8 was chronic low back pain (40.0%; the percentages of "agree" + "strongly agree"), followed by tension-type headache (38.7%), and knee pain due to osteoarthritis (21.8%). Contrastingly, acupuncture was considered far less effective for postoperative nausea/vomiting and prostatitis symptoms (8.3% and 8.7%, respectively). Note that the most common answer was "neither/not sure" for all six conditions listed in Fig 3.

For the question "If acupuncture is recommended in clinical practice guidelines about your conditions or diseases, are you going to try acupuncture?" 34.4% of the respondents answered "agree" or "strongly agree," 22.6% answered "disagree" or "strongly disagree," and 43.0% chose "neither." As to the question on the safety of acupuncture (Q10), 35.6% answered "agree" or "strongly agree", 12.6% answered "disagree" or "strongly disagree", and 51.8% chose "neither".

## Associations between health literacy and experience of or attitudes toward acupuncture

The chi-squared test for cross-tabulation between the efficacy of acupuncture, clinical practice guidelines, or safety of acupuncture and health literacy was found to be significant. Furthermore, the odds ratios between the level of agreement (AA vs. AD/N) and the level of health literacy (HHL vs. LHL) for each category (efficacy, guidelines, and safety) were found to be significant at 3.49, 2.37, and 2.77, respectively. No significant difference was found in the chi-

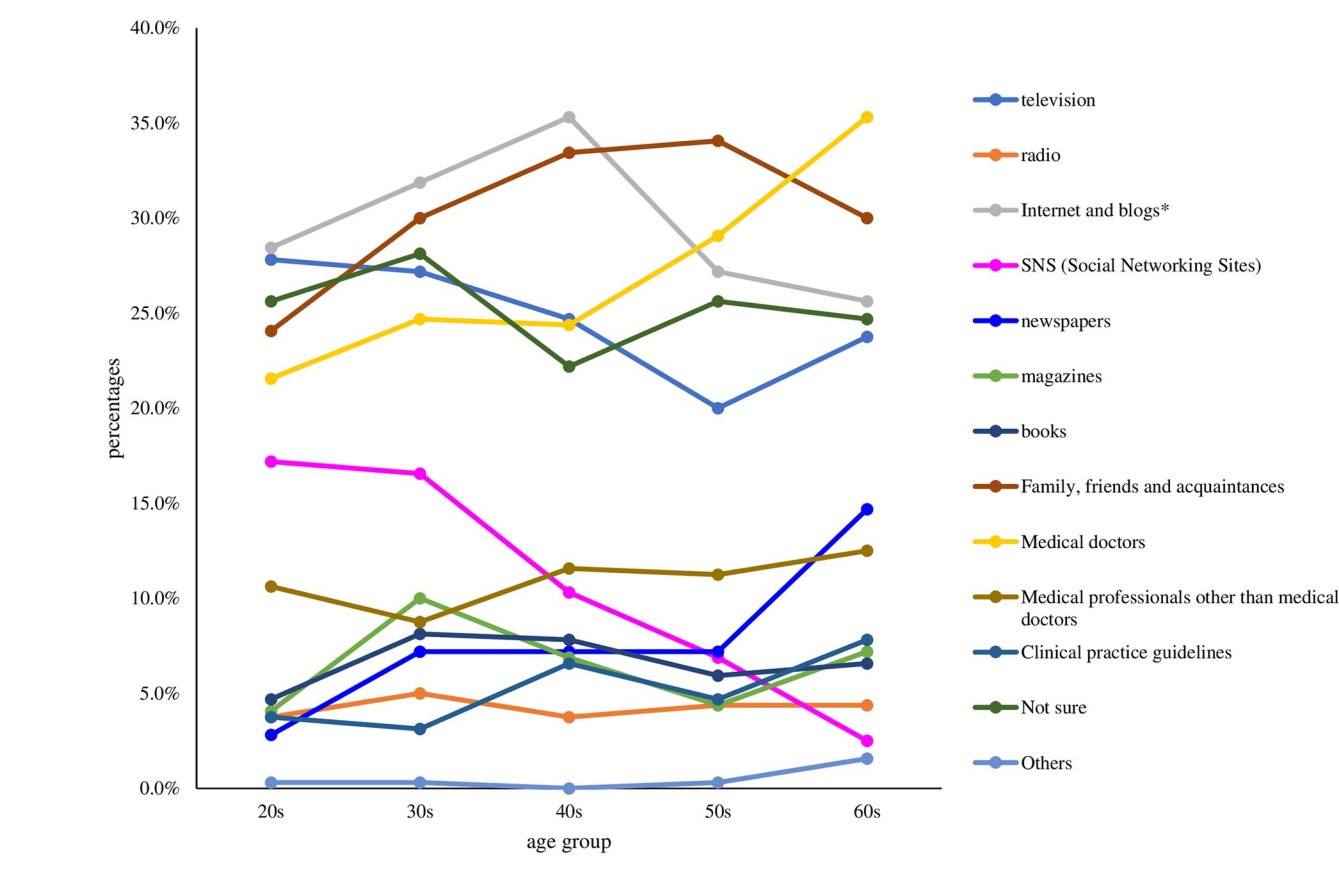

**Fig 2. Sources of information that the respondents rely on deciding whether to choose acupuncture.** Respondents could choose more than one resource option; hence, this figure includes multiple answers. SNS: Social networking sites. * Except for SNS.

squared test and odds ratio for the relationship between health literacy and experience of acupuncture (Table 5).

## Path analysis

Confirming that the fitness indices of the hypothesis model (S1 Fig) were adequate (CFI = 0.968 and RMSEA = 0.038 [95% CI: 0.030–0.048]), the final model was developed after removing the non-significant path, as shown in Fig 4. It was noted that the model fitness indices improved [CFI = 0.978 and RMSEA = 0.031 (95% CI: 0.022–0.040)]. In this model, attitudes toward acupuncture (safety, efficacy, and clinical practice guidelines) were significantly influenced by the participants' health literacy, number of information sources, and experience of acupuncture. However, acupuncture experience was not directly associated with health literacy and health status. However, it should be noted that this model could only explain 3% of health literacy, 5% of efficacy of acupuncture, 10% of possible influence of clinical practice guidelines, and 11% of safety of acupuncture.

## Discussion

To the best of our knowledge, this is the first study to simultaneously question the Japanese population on health literacy items and their experience of and attitudes toward acupuncture.

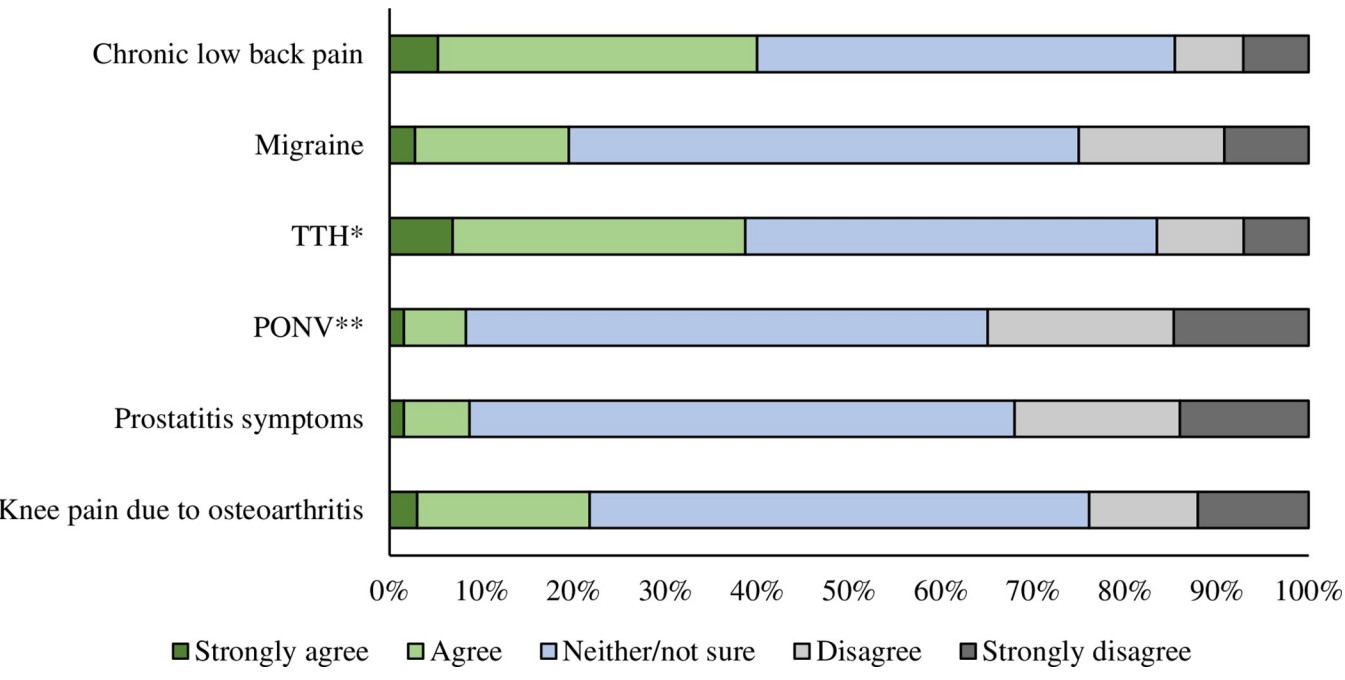

**Fig 3. Degree to which respondents recognize the effectiveness of acupuncture for each condition.** * Tension-type headache (associated with neck and shoulder stiffness). ** Postoperative nausea and vomiting.

The study found that overall, the health literacy of Japanese respondents tended to increase with age, and the main sources of health information differed by generation with respect to SNS, medical doctors, and newspapers. The annual use of acupuncture was 8.2% (95% CI: 6.9–9.6), which was slightly higher than that reported in previous studies, being 6.7% (95% CI: 5.2–8.2) in 2000 [15] and 6.7% (95% CI: 5.2–8.3) in 2005 [16]. Since our survey panel is not proportional to the population of Japan, it is difficult to directly compare previous studies with that of ours. Nevertheless, it is obvious that the annual use of acupuncture in Japan is higher than that in almost all Western countries [35, 36].

**Table 5. Relationship between health literacy and experience of or attitudes toward acupuncture.**

| | | Health Literacy | | | | | | |
|---|---|---|---|---|---|---|---|---|
| | | HHL | LHL | $\chi^2$ | df | p value* | OR | 95% CI |
| Experience of acupuncture | Yes | 213 | 193 | 1.53 | 1 | 0.216 | 1.15 | 0.92–1.44 |
| | No | 584 | 610 | | | | | |
| Efficacy of acupuncture | AA | 155 | 52 | 59.8 | 1 | < .001 | 3.49 | 2.50–4.86 |
| | AD/N | 642 | 751 | | | | | |
| Clinical practice guidelines | AA | 351 | 200 | 64.9 | 1 | < .001 | 2.37 | 1.92–2.93 |
| | AD/N | 446 | 603 | | | | | |
| Safety of acupuncture | AA | 375 | 195 | 90.4 | 1 | < .001 | 2.77 | 2.24–3.43 |
| | AD/N | 422 | 608 | | | | | |

*Pearson's chi-squared test.

HHL: Higher health literacy; LHL: Lower health literacy; df: Degrees of freedom; OR: Odds ratio; CI: Confidence interval; AA: Acupuncture approval; AD/N: Acupuncture disapproval/Neutral

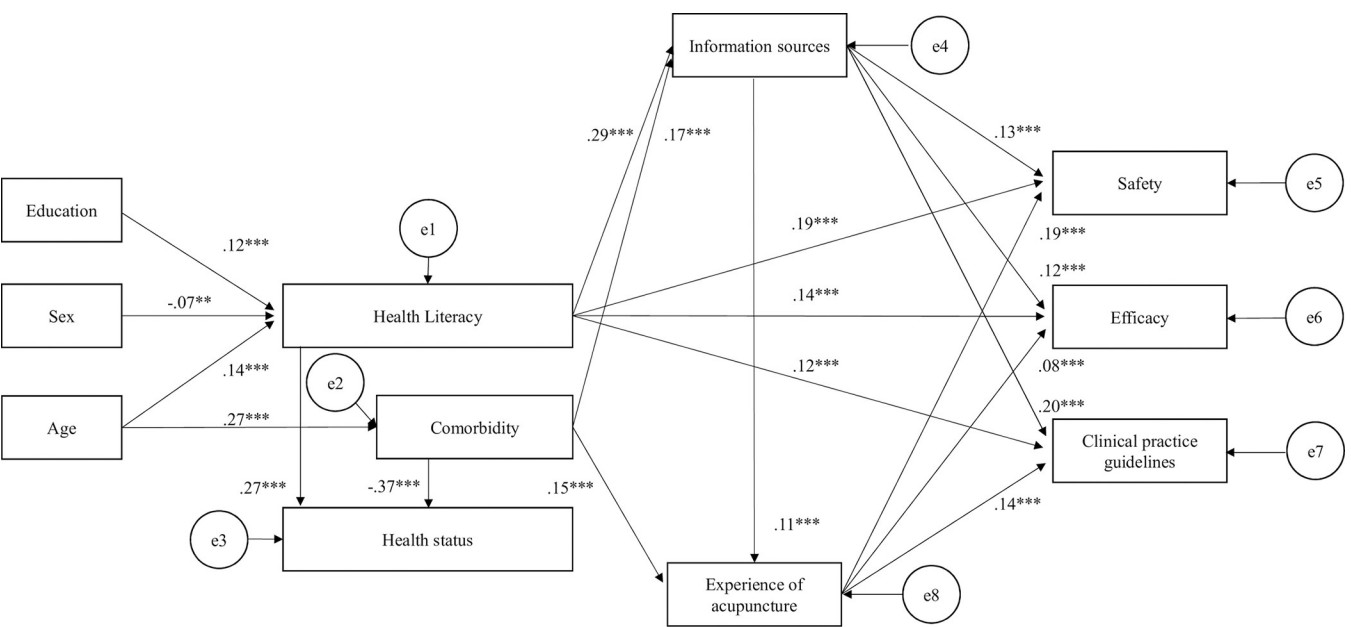

**Fig 4. Path analysis (final model).** The numbers shown correspond to a standardized path coefficient. Model fitness: CFI = 0.978, RMSEA = 0.031 (95% CI: 0.022–0.040). R-squared: health literacy = 0.032, comorbidity = 0.071, health status = 0.201; information sources = 0.117, experience of acupuncture = 0.041, safety = 0.108, efficacy = 0.054; clinical practice guidelines = 0.097. $^*p < 0.05$, $^{**}p < 0.01$, $^{***}p < 0.001$.

In this study, based in Japan, where a relatively larger proportion of the population receives acupuncture than in the West, we focused on whether there is any relationship between health literacy and the choice of acupuncture, along with determining the degree of health literacy of the Japanese population. Thus, this study also aimed to understand whether the Japanese public are making evidence-based choices for acupuncture, the efficacy of which is controversial.

## Health literacy of the Japanese people

Generally, health literacy levels depend on the social, cultural, and demographic background like age, sex, educational attainment, race, marital status, employment status, income, and insurance status of the participants' residential country [13, 37–39]. In this study of Japanese people, we found that age, sex, and educational attainment were associated with health literacy (Fig 4). Specifically, older, educated, and female participants tended to have higher health literacy. These results are not necessarily consistent with those of previous studies conducted in Japan probably due to differences in sample population characteristics [7–10, 17].

We speculate that the differences in information sources among age groups (medical doctors, newspapers, and SNS; Fig 1) may have some influence on the gradual increase in health literacy among age groups. Interestingly, it was found that those in their 60s obtained more information from television and the internet followed by medical doctors and newspapers. Contrastingly, those in their 20s obtained more information from SNS (Fig 1). As shown in Table 3, the odds ratio between HHL and LHL regarding the use of books, internet, and newspapers was particularly high, whereas no significant differences were found for radio and SNS. Considered together, consulting medical doctors and reading newspapers may have some influence on the level of health literacy by age group, although confounding factors that were not included in the question items of this study could also be involved.

It has been highlighted that Japanese people with higher health literacy were more likely to get sufficient health information from multiple sources [10]. The path analysis in this study

also shows the association between health literacy and the number of information sources, supporting previous findings [10]. However, as we discussed above, to raise the level of health literacy of Japanese people, we should consider not only increasing the quantity of health information sources but also scrutinizing the quality and authenticity of each information source and recommend more reliable and effective information sources for each age group.

## Acupuncture and health literacy

This study found that relatively more respondents recognized that acupuncture was effective for chronic low back pain and tension-type headache. However, less than 10% of them thought that this therapy would be effective for alleviating post-operative nausea/vomiting and prostatitis symptoms, indicating a discrepancy between the evidence presented by Cochrane reviews and public perception.

This study's findings are not limited to Japan. Previous studies conducted in other countries have also shown similar results. For example, only 15.5% and 10% of respondents in Australia and China, respectively, thought that acupuncture could reduce nausea and vomiting [40, 41].

Based on the path analysis (Fig 4), it was suggested that Japanese people with higher health literacy are more likely to perceive acupuncture positively. Nevertheless, we could not find a direct relationship between health literacy and acupuncture experience (Table 5, Fig 4). This may be because even respondents with higher health literacy were not provided with sufficient evidence for acupuncture. Thus, it can be said that utilizing potentially effective healthcare measures without being informed about the evidence would be a loss of benefit to the people concerned. Therefore, sufficient evidence and higher health literacy are needed to promote and maintain a better health.

## Healthcare decision-making and health information source

The most common sources of health information that the respondents trusted when deciding whether to use acupuncture were the internet and blogs in their 20s to 40s, family, friends, and acquaintances in their 50s, and medical doctors in their 60s (Fig 2). These results contrast with the findings that television was the most common source of information, except for those in their 40s (Fig 1). Apart from those in their 60s, the choice of acupuncture may be relatively more influenced by personal narratives through internet blogs and family/friends. These information sources do not seem to be appropriate in terms of evidence strength, although patient preferences should be considered.

With respect to the level of evidence, clinical practice guidelines would be an ideal health information source if they were well-produced and trustworthy [42]. In this study, only 34.4% of the respondents chose "agree" or "strongly agree" to the question "if acupuncture is recommended in clinical practice guidelines about your conditions or diseases, are you going to try acupuncture?" and 43.0% chose "neither." Accordingly, we suspect that many respondents did not understand the significance of clinical practice guidelines. Thus, not only does it provide sufficient evidence and appropriate recommendations of clinical practice guidelines, but it is also important to educate people about the concept and role of clinical practice guidelines in the field of healthcare.

However, previous studies have shown that the quality of clinical practice guidelines is not necessarily high, at least in Japan [43]. The recommendations for acupuncture in some Japanese guidelines are also incorrect or methodologically inappropriate [24]. Therefore, the quality of health information sources, including clinical practice guidelines, should be improved in the future.

### Limitations of this study

This study had some limitations. First, because the number of respondents in each age and sex group was 160, it was not proportional to the population structure of Japan. Second, the respondents were not selected through random sampling, which is a common issue in web-based surveys using panel respondents. Third, it is difficult to know the classification and reasons of those who agreed to respond but were not included among the respondents (1,692 in this study), a limitation that is inherent to panel-based studies. Fourth, we did not collect data on the respondents' income, which may have influenced the path analysis, particularly in terms of education, health literacy, and experience of acupuncture. Fifth, we did not focus on the relationship and influence of the type of comorbidities because the respondents were not proportional to the Japanese population, as mentioned above, and morbidity was less prevalent than we expected. Sixth, for the 25.4% of respondents who had received acupuncture in their lifetime, their attitudes toward this therapy might have been influenced by actual positive or negative experiences, rather than their health literacy. However, detailed questions on how those who received acupuncture in the past felt about the treatment were not included in this survey. Seventh, the measurement of health literacy was based on self-reporting, which may introduce biases, such as recall bias and over- or underestimation.

Despite these limitations, this is the first study to assess the relationship between health literacy and attitudes toward acupuncture, and it provides useful suggestions for the future of health education.

### Conclusions

In this study, involving a panel of 1,600 respondents stratified by age and sex, we found that age, educational attainment, and sex might be associated with the health literacy of Japanese people. Older, educated, female participants tended to have higher health literacy. Furthermore, it can be deduced that consulting medical doctors and reading newspapers may have influenced the level of participants' health literacy. Although Japanese people with higher health literacy are more likely to perceive acupuncture positively, they do not necessarily have sufficient knowledge of the clinical evidence of acupuncture, and their decision to receive acupuncture may be more dependent on personal narratives from their family and acquaintances rather than clinical research evidence.

A challenge that needs to be addressed in the future is individual education on how to choose a reliable health information source and organizational efforts to provide more reliable information. This would also be true for complementary and integrative healthcare approaches that are controversial regarding efficacy and safety, for the sake of better-informed health decision-making.

### Supporting information

**S1 Table. Checklist of the Strengthening the Reporting of Observational Studies in Epidemiology (STROBE) statement [26].**
(DOCX)

**S2 Table. Checklist for Reporting Results of Internet E-Surveys (CHERRIES) [27].**
(DOCX)

**S3 Table. Checklist of the reporting guidelines for structural equation modeling [28].**
(DOCX)

**S4 Table. 11-item questionnaire.** This questionnaire comprises four categories of health status (Q1, 2), health literacy (Q3, 4), experience of receiving acupuncture (Q5, 6) and recognition and choice behavior about acupuncture (Q7–11). Regarding the health literacy measurement of Q3, we used a 5-item questionnaire developed by Ishikawa et al. [7]. (DOCX)

**S1 Fig. Path analysis (hypothesis model).** Rectangles are the observed variables. The values on the single-headed arrows are standardized regression weights. Model fitness: CFI = 0.968, RMSEA = 0.038 (95% CI 0.030–0.048). $^*p < 0.05$, $^{**}p < 0.01$, $^{***}p < 0.001$. (TIF)

## Author Contributions

**Conceptualization:** Yuse Okawa, Hitoshi Yamashita.

**Data curation:** Yuse Okawa.

**Formal analysis:** Yuse Okawa, Norio Ideguchi.

**Funding acquisition:** Yuse Okawa.

**Investigation:** Yuse Okawa.

**Methodology:** Yuse Okawa.

**Project administration:** Yuse Okawa.

**Supervision:** Hitoshi Yamashita.

**Validation:** Norio Ideguchi.

**Visualization:** Yuse Okawa.

**Writing – original draft:** Yuse Okawa.

**Writing – review & editing:** Norio Ideguchi, Hitoshi Yamashita.

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
