## [Decision Letter · Decision Letter 0]

9 Feb 2023

PONE-D-22-29061Japanese people’s health literacy and their attitudes toward acupuncture: A web-based cross-sectional surveyPLOS ONE

Dear Dr. Okawa,

Thank you for submitting your manuscript to PLOS ONE. After careful consideration, we feel that it has merit but does not fully meet PLOS ONE’s publication criteria as it currently stands. Therefore, we invite you to submit a revised version of the manuscript that addresses the points raised during the review process.

We look forward to receiving your revised manuscript.

Kind regards,

Zsombor Zrubka, PhD

Academic Editor

PLOS ONE

Journal Requirements:

Additional Editor Comments (if provided):

Dear Authors,

the reviewers have sent their comments. In addition, I suggest the following:

a) Please use an adequate reporting guides for your study: e.g., CHERRIES for online surveys and Todd 2017 (doi: 10.4236/psych.2017.89086 ) for SEM.

b) Please address the following issues concerning the data analysis:

- neither / not sure responses were categorised as acupuncture disapproval. However, this option may also indicate low familiarity with acupuncture, and naturally be associated with low health literacy. As such, the conclusions of the current analysis are potentially misleading.

- actual positive/negative experience with acupuncture may override the beliefs driven by health literacy. The authors should explore this possible effect and find ways to reflect in the analysis.

- using health literacy as a continuous predictor may be more adequate than a dichotomised item

c) The applied SEM was confirmatory SEM, which requires the advance statement of a hypothesis (e.g. pre-specified model), and fit indices then indicate whether the hypothesis is plausible.

In the context of this research, I suggest considering if exploratory SEM (eg. pls-SEM) would be more adequate than CB-SEM. (e.g. Hair 2019, doi 10.1108/EBR-11-2018-0203)

In the current form, the analysis of results is potentially misleading.

Thank you,

Your academic editor.

Reviewers' comments:

Reviewer's Responses to Questions

**Comments to the Author**

1. Is the manuscript technically sound, and do the data support the conclusions?

Reviewer #1: Yes

Reviewer #2: Partly

2. Has the statistical analysis been performed appropriately and rigorously? 

Reviewer #1: Yes

Reviewer #2: Yes

3. Have the authors made all data underlying the findings in their manuscript fully available?

Reviewer #1: Yes

Reviewer #2: Yes

4. Is the manuscript presented in an intelligible fashion and written in standard English?

Reviewer #1: Yes

Reviewer #2: Yes

5. Review Comments to the Author

Reviewer #1: 1) Intro section: It is too short overall -- this section needs to be profound. It is vital to state the objective of the study clearly

2) There needs to be more information regarding the association between health literacy and attitudes toward acupuncture.-> line 62,63,64. I have seen various articles regarding health literacy and CAM; however, this is the first time linking health literacy to acupuncture. It is essential to mention why.

Do people not use acupuncture due to health literacy in Japan? Is there an article or statistics associated with it?

3) Please include a web-link for Mellinks.

4) Please indicate how Mellinks gather panels in the Methods section.

5) Please indicate any rewards given to the panel and clearly state how you have recruited panels step-by-step.

6) Are there no dropouts from 1600 people?

7) Are there any discrepancies among the university staff members in reviewing the survey questions?

8) Have you thought of performing a logistic regression analysis?

Reviewer #2: To my understanding, the research question of this study is twofold. Firstly, this study examines the relationship between health literacy and the choice of acupuncture. It also aims to understand the extent to which the respondents' decision to use acupuncture depends on their knowledge and understanding of reliable sources. It also gives insight of the degree of health literacy of the Japanese population (based on a not representative sample).

Other studies have been done on the relationship between the use of alternative and complementary medicine and health literacy, including those conducted among Japanese population. The novelty of the study is that it is the first to examine the relationship between the attitude to use acupuncture and health literacy among Japanese respondents. Narrowing the scope of complementary treatments to acupuncture could be justified by, that acupuncture is more widespread among the Japanese population than elsewhere in the world, so it may be relevant to study it in the context of health literacy. I would suggest that the results of previous research on a similar topic also be mentioned in the introduction.

Used methods and reporting quality:

To answer the research question, authors conducted an online survey among 1600 Japanese respondents. Respondents were asked based on four categories: health status, health literacy, experience with acupuncture, and recognition and choice behaviour regarding acupuncture.

To measure health literacy, they used a 5-item questionnaire developed by Ishikawa et al. The justification for the use of that scale seems incomplete. The authors have referred to several known measures, choosing one that has been piloted on a small sample size in a small circle (office workers). In this case, I think it is crucial to explain and justify the choice of the scale.

The authors developed category variables from the health literacy scale scores. Respondents who scored above the group median were classified as high health literate, and those who scored below the group median were classified as low health literate. I think this is a good solution, but I would emphasise in the results that high and low health literacy should be understood in relation to the scores of the sample.

Cross tabulation and chi-squares were used to examine the relationships between clusters, which are the appropriate statistical methods for the sample, and their results were reported in a clear and transparent way.

All scales used were based on self-report, a potential weakness of the study that is not addressed in the weaknesses section or in the description of the measurement scales.

For the descriptive statistical analysis of the data I would suggest the use of visualization tools (for example pie charts for describing the percentages of people self-perceived to be healthy, not healthy or for example the use of histograms to represent HL points distribution.

Both the conclusion and the title of the paper refer to a study of the attitudes of Japanese people, but the sample was not representative (as the authors pointed out transparently), so I think it would be more accurate to limit the conclusions to the sample rather than to Japanese people

One recommendation regarding the format is that I would suggest the authors formatting the tables in a standard format.

Overall, in my opinion the research question has been answered with appropriate statistical tests and the quality of the reporting is is good. I therefore propose the acceptance of the paper with minor adjustments.

6. PLOS authors have the option to publish the peer review history of their article (what does this mean?). If published, this will include your full peer review and any attached files.

Reviewer #1: No

Reviewer #2: No

---

## [Author Response · Author response to Decision Letter 0]

25 Mar 2023

See the attached file named "Response to Reviewers."

---

## [Decision Letter · Decision Letter 1]

26 Jun 2023

PONE-D-22-29061R1Japanese people’s health literacy and their attitudes toward acupuncture: A web-based cross-sectional survey with a panel of residents in JapanPLOS ONE

Dear Dr. Okawa,

Thank you for submitting your manuscript to PLOS ONE. After careful consideration, we feel that it has merit but does not fully meet PLOS ONE’s publication criteria as it currently stands. Therefore, we invite you to submit a revised version of the manuscript that addresses the points raised during the review process.

We look forward to receiving your revised manuscript.

Kind regards,

Huijuan Cao, Ph.D.

Academic Editor

PLOS ONE

Reviewers' comments:

Reviewer's Responses to Questions

**Comments to the Author**

1. If the authors have adequately addressed your comments raised in a previous round of review and you feel that this manuscript is now acceptable for publication, you may indicate that here to bypass the “Comments to the Author” section, enter your conflict of interest statement in the “Confidential to Editor” section, and submit your "Accept" recommendation.

Reviewer #1: All comments have been addressed

Reviewer #3: (No Response)

2. Is the manuscript technically sound, and do the data support the conclusions?

Reviewer #1: Yes

Reviewer #3: (No Response)

3. Has the statistical analysis been performed appropriately and rigorously? 

Reviewer #1: Yes

Reviewer #3: (No Response)

4. Have the authors made all data underlying the findings in their manuscript fully available?

Reviewer #1: Yes

Reviewer #3: (No Response)

5. Is the manuscript presented in an intelligible fashion and written in standard English?

Reviewer #1: No

Reviewer #3: (No Response)

6. Review Comments to the Author

Reviewer #1: All comments have been addressed; however, I would like to stress out that this manuscript needs to get professional editing and proofreading services.

Reviewer #3: This is an excellent study and the authors have revised the manuscript in response to comments made by other reviewers. However, the authors need to make further revisions and improvements to address the following points.

* The background needs to continue to be refined to explain the need for the study.

*Page 5, Line 66-68: Why did the authors make this assumption? Based on the previous description, it is difficult to think of this assumption.

*Page 8, Line121-123: In the background of the study, the authors primarily wanted to address the correlation between health literacy (Education Level, line 76-84) and people's attitudes toward acupuncture, so why did the authors not group people according to education level but rather according to age? Here the authors need to provide an explanation and cite evidence to justify the need to do so.

*Page 8, line 129: What were the main areas of pre-work or reference on which the questionnaire was developed? Who were the people involved in the development of the questionnaire? What are the backgrounds of each of these people? The authors need to elaborate.

*Page 13, line 215: Therefore, the completion rate was of 48.6%.

A completion rate (effective rate) of 80% for the questionnaire would be considered feasible for implementation. However, 48.6% was reported here. Have the authors considered the reasons for this result?

7. PLOS authors have the option to publish the peer review history of their article (what does this mean?). If published, this will include your full peer review and any attached files.

Reviewer #1: No

Reviewer #3: No

---

## [Author Response · Author response to Decision Letter 1]

27 Jul 2023

We have attached the response to reviewers as a separate file.

---

## [Decision Letter · Decision Letter 2]

16 Aug 2023

PONE-D-22-29061R2Japanese people’s health literacy and their attitudes toward acupuncture: A web-based cross-sectional survey with a panel of Japanese residentsPLOS ONE

Dear Dr. Okawa,

Thank you for submitting your manuscript to PLOS ONE. After careful consideration, we feel that it has merit but does not fully meet PLOS ONE’s publication criteria as it currently stands. Therefore, we invite you to submit a revised version of the manuscript that addresses the points raised during the review process.

We look forward to receiving your revised manuscript.

Kind regards,

Patricia de Moraes Mello Boccolini

Academic Editor

PLOS ONE

Additional Editor Comments:

In the Introduction: The authors must expound more clearly on this study's precise objectives. It would be prudent to incorporate a more comprehensive presentation of studies concerning health literacy and acupuncture, not solely confined to Japan but also encompassing nations where acupuncture holds a notable prevalence.

Within the Methodology: The study engaged 1600 participants. The participant selection procedure demands more excellent elucidation, accompanied by articulating any instances of participant attrition. It is imperative to consider the integration of additional statistical analyses, such as logistic regression.

Regarding the Discussion: The authors acknowledge that the study's participant cohort does not exhibit a representative cross-section of the broader Japanese populace. Consequently, it is recommended to revisit the formulation of the article title to reflect this limitation accurately.

Reviewers' comments:

Reviewer's Responses to Questions

**Comments to the Author**

1. If the authors have adequately addressed your comments raised in a previous round of review and you feel that this manuscript is now acceptable for publication, you may indicate that here to bypass the “Comments to the Author” section, enter your conflict of interest statement in the “Confidential to Editor” section, and submit your "Accept" recommendation.

Reviewer #1: All comments have been addressed

Reviewer #3: (No Response)

2. Is the manuscript technically sound, and do the data support the conclusions?

Reviewer #1: Yes

Reviewer #3: (No Response)

3. Has the statistical analysis been performed appropriately and rigorously? 

Reviewer #1: Yes

Reviewer #3: (No Response)

4. Have the authors made all data underlying the findings in their manuscript fully available?

Reviewer #1: Yes

Reviewer #3: (No Response)

5. Is the manuscript presented in an intelligible fashion and written in standard English?

Reviewer #1: Yes

Reviewer #3: (No Response)

6. Review Comments to the Author

Reviewer #1: It is a very minor issue; however, I would like to suggest changing the title to be in the form of a research-article.

Please review below:

https://blog.wordvice.com/how-to-write-the-perfect-title-for-your-research-paper/

Reviewer #3: (No Response)

7. PLOS authors have the option to publish the peer review history of their article (what does this mean?). If published, this will include your full peer review and any attached files.

Reviewer #1: No

Reviewer #3: No

---

## [Author Response · Author response to Decision Letter 2]

21 Sep 2023

We have attached the response to reviewers as a separate file.

---

## [Editor Report · Decision Letter 3]

28 Sep 2023

Relationship between health literacy and attitudes toward acupuncture: A web-based cross-sectional survey with a panel of Japanese residents

PONE-D-22-29061R3

Dear Dr. Okawa,

We’re pleased to inform you that your manuscript has been judged scientifically suitable for publication and will be formally accepted for publication once it meets all outstanding technical requirements.

Kind regards,

Patricia de Moraes Mello Boccolini

Academic Editor

PLOS ONE
---

## [Editor Report · Acceptance letter]

13 Oct 2023

PONE-D-22-29061R3 

Relationship between health literacy and attitudes toward acupuncture: A web-based cross-sectional survey with a panel of Japanese residents 

Dear Dr. Okawa:

I'm pleased to inform you that your manuscript has been deemed suitable for publication in PLOS ONE. Congratulations! Your manuscript is now with our production department. 

Kind regards, 

on behalf of

Dr. Patricia de Moraes Mello Boccolini 

Academic Editor

PLOS ONE